# ViT-MoQ: Revisiting Momentum Queues for Resource-Efficient Vision Transformers and Domain Generalization in Self-Supervised Learning

## Abstract

Self-supervised learning (SSL) has achieved remarkable success in computer vision, but current state-of-the-art methods require substantial computational resources with large batch sizes (4096) and multi-GPU clusters. We present ViT-MoQ, a compute-efficient contrastive SSL method that reintroduces momentum queues to Vision Transformer architectures. Our key insight is that symmetric encoder architectures are essential for queue-based learning in ViTs, contrary to the asymmetric designs prevalent in recent SSL methods. ViT-MoQ achieves competitive performance while requiring only a single consumer GPU, considerably reducing compute requirements. On ImageNet-1K linear probing, ViT-MoQ achieves competitive performance on as few as 165 GPU hours. More interestingly, we show superior domain generalization capabilities: when trained on DomainNet-Real, ViT-MoQ significantly outperforms MoCo variants across all tested domains (e.g., 44.4% vs 28.4% on painting, 44.81% vs 0.6% on quickdraw). Our work challenges the assumption that momentum queues are obsolete in the transformer era and demonstrates that architectural compatibility, not inherent limitations, was the barrier to their adoption. ViT-MoQ enables more SSL applications by making high-quality self-supervised learning accessible on modest hardware while learning more transferable, domain-agnostic representations and enabling sustainable, green AI research practices. Code will be published.

## 1 Introduction

Self-Supervised Learning (SSL) has perceived much attention, especially in the language domain. The advent of models such as GPT (Yenduri et al., 2023), BERT (Devlin et al., 2019), and RoBERTa (Liu et al., 2019) demonstrates unprecedented capabilities and reasoning abilities. Early contrastive methods like MoCo-v1 (He et al., 2020) or SimCLR (Chen et al., 2020a) paved the way for self-supervised learning. Now, current methods span contrastive approaches like DINO v3 (Siméoni et al., 2025), SwAV (Caron et al., 2021a), masked autoencoding (MAE He et al. (2021), iBOT Zhou et al. (2022)) and hybrids such as EsViT (Li et al., 2022a) and BEiT (Bao et al., 2022). These methods achieve state-of-the-art results, often beating their supervised counterparts. Although truly remarkable in accuracy and achievement, most of these methods require a large compute budget, often employing clusters of A100 and H100 to achieve these numbers. This high computation requirement often arises from large batch sizes. These large batch sizes (around or more than 4096) facilitate a sufficiently diverse group of negative samples to draw from. There have been multiple attempts to combat this, using clever sampling techniques like (Tan et al., 2023) or using completely different small batch methods, such as TriBYOL (Li et al., 2022b), that learns to stabilize on small batch sizes using triplet loss. Other methods like S3L (Cao & Wu, 2021) completely scale down the size of SSL using small architectures, small datasets, and small resolutions. Other methods like SimSiam (Chen & He, 2020) use small batch sizes (512) but also use large 8-GPU compute clusters. These innovations highlight the feasibility and need for high performance, robust, and efficient self-supervised learning in a low compute setting. Based on this background, it is also clear that there is a need for a reliable SSL method which gives competitive performance on low compute and/or low batch sizes. Currently, SSL's computational demands limit deployment in resource-constrained

environments. Building on the "starting small" principle by Elman (1993), we show that efficiency constraints can even improve representation learning through optimal architectural design.

Despite the impressive performance of these SSL methods, a limitation emerges when these methods are deployed on different domains. Domain generalization remains a significant challenge for self supervised learning. Generalizing to completely unseen domains is also very heavily dependent upon lighting conditions, abstractions and visual characteristics. This limitation is particularly pronounced in low-compute SSL scenarios, where the reduced batch sizes and limited negative sampling may lead to representations that are overly specialized to the training domain. Another challenge for recent SSL methods, is the asymmetric encoder architecture with in-batch sampling. This might hurt the learning of domain-agnostic features, due to a limited number of negatives samples. The high computational requirements of existing methods not only limit accessibility but also constrain the diversity of training data that can be processed, potentially reducing the model's exposure to domain variations that would improve generalization.

To this end, we present ViT-MoQ: ViT-MoQ addresses two critical gaps in modern self-supervised learning. First, it demonstrates that robust and transferable representations can be learned efficiently on a single consumer-grade GPU (RTX 4090, 24GB VRAM) using a momentum queue with a symmetric encoder, significantly reducing compute and batch size requirements compared to previous ViT-based SSL approaches. Second, ViT-MoQ exhibits strong domain generalization, showing that these representations transfer effectively across diverse datasets — an area that remains largely underexplored in the literature. Before the era of in-batch negative sampling and large batch sizes for stability, The Fundamental AI Research lab proposed MoCo: a novel framework to decouple the batch size from the pool of negative samples by introducing a memory queue and an EMA key encoder. MoCo-v1 (He et al., 2020) and MoCo-v2 (Chen et al., 2020b) demonstrated substantial success, achieving competitive self-supervised learning results and notably outperforming supervised pre-training on downstream detection and segmentation tasks. However, for MoCo-v3 (Chen et al., 2021), the authors decided to drop the queue in favour of in-batch sampling, claiming diminishing returns for large batch sizes. Subsequently, the memory queue seems to be vanishing from current literature, with methods moving more in favour of in-batch negative sampling. ViT-MoQ aims to address this and reintroduces the momentum queue to a ViT (Dosovitskiy et al., 2021) backbone in a contrastive learning setup. Our work not only shows that stable SSL learning with a momentum queue is possible and can achieve competitive results, but also that a queue mechanism can significantly improve domain generalization capabilities.

## 2 RELATED WORK

Self-Supervised Learning has advanced quickly from foundational approaches like MoCo-v1/v2 (He et al., 2020; Chen et al., 2020b) and SimCLR (Chen et al., 2020a). These approaches laid the foundation for further works like DINO v1,v2,v3 (Caron et al., 2021b; Oquab et al., 2023; Siméoni et al., 2025) and MAE (He et al., 2021) and dominate the field with never-before-seen state-of-the-art accuracy on Imagenet-1k. These methods move beyond just contrastive losses and employ techniques like student-teacher networks and knowledge distillation or pixel reconstruction. However, this accuracy comes at a high computational cost, usually requiring clusters of powerful GPUs and hundreds of hours of GPU training. This reliance on high and expensive computing is at odds with the principles highlighed in Green AI (Schwartz et al., 2020), which argue that efficient, low compute methods are critical for sustainable AI. To counter this, there have been several attempts to reduce the batch size and/or compute dependency.

One of the most influential methods is BYOL (Grill et al., 2020). BYOL shows that contrastive learning is possible without explicit negative pair mining. Traditionally, BYOL works with a batch size of 4096 with a distributed load over a 512 TPU cluster. However it can be scaled down to a batch size of 512 on a 64 TPU cluster and training times of over four days. Another work which extends the idea of contrastive learning without negative sample mining is SwAV (Caron et al., 2021a). SwAV reduces the batch size substantially, using a batch size of 256, and can be run on four GPUs. Another extension of the BYOL method is TRIBYOL, one of the most computationally efficient and less GPU-intensive methods. The authors combine a network with a triplet loss along with the BYOL framework. This method works with batch sizes of less than 128 on a single GPU. However, the measured metrics are taken from CIFAR-10, CIFAR-100, and MNIST datasets. More representative

ImageNet-1k or ImageNet-100 Linear Probes are missing. MoBy (Xie et al., 2021) is another variant which combines the queue and EMA from MoCo-v2 and negative sample mining from BYOL. MoBY uses the swin transformer (Liu et al., 2021) as a backbone and adopts an asymmetrical encoder architecture (a projection head and a prediction head on the query encoder, whereas just a projection head on the key encoder). MoBY's official codebase is implemented in a distributed data parallel (DDP) fashion, suggesting multi-GPU usage.

Other methods include Barlow Twins (Zbontar et al., 2021), which adds an empirical cross-correlation matrix to avoid collapse. Usually trained on a batch size of 2048, with 32 V100 GPUs over 8 hours, it is possible to scale this method down to a batch size of 256 with minimal loss in performance. VICReg (Bardes et al., 2022) is another innovative method for compute-restrained SSL. They achieve stability and performance by applying two regularization terms over the projection head space. VICReg uses a batch size of 256. However, they did not publish their GPU compute requirements. FastSiam (Pototzky et al., 2022) tries to replicate ImageNet weights of SimSiam with as little computational requirements as possible and using small batch sizes of 32. But most of their downstream tasks only focus on the COCO dataset.

While domain generalization of supervised models has been extensively studied (Zhou et al., 2023), most SSL literature evaluates representation quality via linear probing on the same dataset (e.g., ImageNet), with limited analysis of cross-domain robustness. Some approaches have shown emergent generalization properties; for instance, DINO (Caron et al., 2021b) exhibits strong features for semantic segmentation across domains. Methods like Domain-Agnostic Approach to Contrastive Learning (DACL) (Verma et al., 2021) and SelfReg (Kim et al., 2021) introduce explicit regularization losses to learn domain-invariant features. However, these approaches often maintain the high computational costs of standard SSL, rely on Imagenet pretraining, or use multi-domain training. Domain generalization with SSL is an extremely underexplored area, with barely any baselines or standards for comparison. Lack of standardized evaluation protocols such as Single Source Domain Generalization (SSDG) or Leave-one-Group-out (LOGO) further exacerbates this problem.

From this overview, it is clear that even the so-called "small batch" or "low compute" methods in SSL often rely on multi-GPU setups and batch sizes of at least 512. Moreover, almost all such approaches default to ResNet-50 backbones, leaving transformer-based self-supervised learning underexplored in resource-constrained settings. ViT-MoQ aims to address all these points by reintroducing the momentum queue to a ViT backbone with a symmetric encoder setup, demonstrating that robust and transferable domain-agnostic representations can be learned efficiently on a single consumer-grade GPU (RTX 4090, 24GB VRAM).

## 3 METHOD

We demonstrate that symmetric encoder architectures are essential for stable momentum queue training in Vision Transformers. Unlike previous SSL methods that employ asymmetric designs (prediction head on query encoder only), ViT-MoQ uses identical projection-only architectures for both query and key encoders, resolving the fundamental incompatibility that prevented successful queue-transformer integration. Our method revisits the momentum queue from the MoCo-v2 architecture and a ViT-S/16 as backbone for feature encoding. Both encoders consist of the ViT followed by a single projection head. Critically, we eliminate the prediction head used in asymmetric SSL methods, as our analysis shows this creates representation space mismatches that destabilize queue-based training in transformers. We follow the process of positive and negative pair augmentation. Our augmentation policy is based on Wu et al. (2018) and includes random resized cropping, color jitter, random gray scale, GaussianBlur, solarization, posterization, and random horizontal flip. The objective is to maximize the agreement between the positive and negative views of the same image and minimize the agreement between the positive view and all other views in the queue. A contrastive loss Hadsell et al. (2006) achieves this by giving a high value when the query image $q$ and corresponding key image $k_+$ have a high similarity score. We use the standard InfoNCE loss by van den Oord et al. (2019).

$$L_q = -\log \frac{exp(q.k_+/\tau)}{\sum exp(q.k_i/\tau)} \tag{1}$$

where $\tau$ is the temperature parameter. The update for the key encoder is done using an exponential moving average following the equation

$$\theta_k \leftarrow m * \theta_k + (1 - m) * \theta_q \qquad (2)$$

where $\theta_q$ and $\theta_k$ are the parameters of the query and key encoders, respectively.

Figure 1 explains the architecture in detail. The main motivation to use a queue here is to decouple

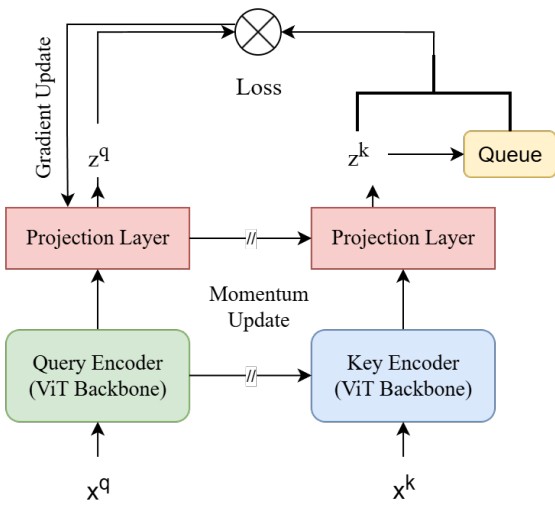

Figure 1: ViT-MoQ architecture diagram

the batch size from the negative sample set. Within a queue, all samples present act as negative samples, as opposed to in-batch sampling. This should lead to more diverse and richer negatives and in more robust feature learning.

Most SSL architectures use an asymmetrical encoder architecture i.e a projection layer and a prediction layer on the query encoder and just a projection layer on the key encoder. However, in our study we found that the inclusion of the prediction layer is incompatible with the training set up. We conducted an ablation study with and without the prediction layer and in all cases the model with the prediction layer did not learn properly, the loss did not decrease and the downstream performance was not optimal. Hence, for all our evaluations we removed the prediction head from the architecture, leaving just a projection head on both the query and key encoder. The projection layer is a small MLP with one hidden layer and ReLU as activation function.

In order to focus on the low resource aspect, all ImageNet-100 experiments are carried out on a single RTX 4090 and with a batch size of 256. This consumes an average of 18GB of VRAM. ImageNet-1k experiments required 20GB of VRAM and a batch size of 512. We use mixed precision training for ImageNet-1k, which allows for the larger batch size. We use full precision training for ImageNet-100.

## 4 EXPERIMENTS

To test the model, we used three different datasets and performed an ablation study as well as a queue size parameter exploration study. We measured downstream performance on a linear probe with a frozen backbone on ImageNet-1k and ImageNet-100. Additionally, we analyzed the robustness of the feature representation by training on Domainnet-Real and testing the frozen backbone on all different domains. In line with the authors of MoCo-v3, we also observed instability during training. This training instability manifests as degradation of downstream performance rather than catastrophic failure. We employ the same patch freezing trick as Chen et al. (2021) to improve stability and downstream performance. The Following data sets were used:

| Method | Architecture | Top-1 | GPU Hours |
|---|---|---|---|
| ViT-MoQ (ours) | ViT-S/16 | 61.3 | 165 |
| MoCo-v1 | ResNet-50 | 68.6 | 424 |
| MoCo-v2 | ResNet-50 | 71,1 | 424 |
| MoCo-v3 | ViT-S/16 | 72.5 | 614 |

Table 1: ViT-MoQ achieves similar accuracy with fewer GPU hours compared to MoCo-v3, showing efficiency advantages. Moreover, we use the lowest GPU hours despite having a transformer backbone

ImageNet-1k (Deng et al., 2009) has 1000 classes in the dataset for image classification. The training set has around 1.28M images, and the validation set has 50k.

ImageNet-100 is a subset of ImageNet-1k with 100 classes. The dataset has around 170k images in total. 135k were used for training and the rest for testing and validation.

Domainnet Peng et al. (2019) is a collection of 6 separate datasets, each with a different domain but the same labels. The domains are Real, Clipart, Infographic, Quickdraw, Painting and Sketch. Every domain has a different number of images.

## 4.1 COMPUTE EFFICIENCY

We trained the model with AdamW optimizer, a learning rate of $3e-4$ and a cosine learning scheduler with a warm up of 10 epochs. The $\tau$ is set to 0.07, momentum parameter m to 0.999, and queue size k to 131072. We limit training epochs to 400 epochs. We also employ mixed precision training which allows for a batch size of 512 on the RTX 4090 using a GPU VRAM of 20GB. Our choice of parameters is influenced by prior works like MoCo-v2/v3, SimCLR. We choose the queue size to be approximately 10% of the training data size.

On ImageNet-1k, as per the linear probe protocol, we freeze the ViT backbone, remove the projection layer, and train a classifier head (output dims=1000). We can report a top-1 accuracy of 61.3% without a hyperparameter search and no specific fine-tuning of the architecture for this dataset. While not being able to completely reproduce state-of-the-art downstream performance this way, a clear gain in compute efficiency is evident from plots 2 and 3, which was the main focus of our study.

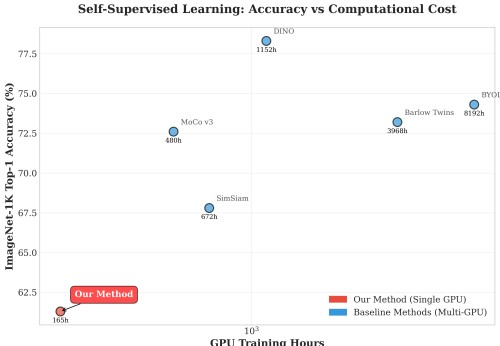

Figure 2: ImageNet-1k LP Accuracy vs GPU Hours for known SSL methods

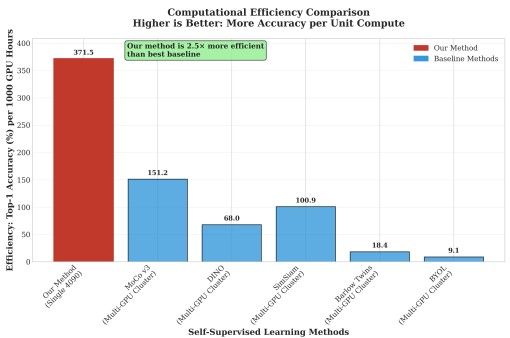

Figure 3: ImageNet-1k LP Accuracy per 1000 GPU Hours for known SSL methods

From figure 2 and figure 3, we can see that ViT-MoQ offers a perfect balance of performance against computation costs. ViT-MoQ offers twice the efficiency of MoCo-v3 and around 49x the efficiency of BYOL. This is a massive reduction in compute power without significantly sacrificing downstream performance. ViT-MoQ also shows that large queue sizes (131k) are stable and do not cause failure in training. ViT-MoQ achieves 80% of SOTA performance using only 2-3% of the computational resources, demonstrating remarkable efficiency without extensive hyperparameter optimization. This suggests significant potential for further performance gains through systematic tuning

| Model | clipart | infographic | painting | quickdraw | real | sketch (%) |
|---|---|---|---|---|---|---|
| MoCo-v2 LP | 16.7 | 7.3 | 28.4 | 0.6 | 33.6 | 18.4 |
| MoCo-v3 LP | 29.3 | 13.9 | 39.3 | 2.2 | 43.2 | 27.5 |
| ViT-MoQ LP (ours) | 48.9 | 23.2 | 44.4 | 44.81 | 65.1 | 36.2 |

Table 2: Domain Generalization results of MoCo variants and ViT-MoQ

while maintaining our efficiency advantages. A core challenge in low-compute SSL is direct comparison with other methods due to lack of standardized benchmarks. VICReg, a leading small batch contrastive method, reports a top-1 accuracy of 73% on ImageNet-1k. While they report a small batch size of 256, there is no report of the required computational power. Moreover, VICReg uses a ResNet-50 backbone. Estimating the performance of VICReg on a ViT backbone on a single GPU is a non-trivial task. Methods like TriBYOL do not report ImageNet-1k or ImageNet-100 top-1 accuracies, but focus more on the smaller CIFAR-10, CIFAR-100 and MNIST datasets. FastSiam works on a single GPU in small batch sizes (128), but focuses on comparing its performance with ImageNet weights on downstream COCO classification tasks. ViT-MoQ establishes the first comprehensive baseline for resource-constrained ViT-based SSL on ImageNet-1K, filling a critical gap where existing efficient methods rely primarily on ResNet architectures. Our work provides essential benchmarks for future research in transformer-based efficient SSL.

## 4.2 Domain Generalization

To test the domain generalization performance of our method, we trained on DomainNet-Real. DomainNet-Real has 345 classes and unlike ImageNet, DomainNet-Real is not a curated dataset and shows class imbalance and a certain variance in label noise. This makes it the perfect testbed to evaluate the robustness and domain agnosticism of the learned representations. We use the AdamW optimizer and a parameter set of $queue = 16384, \tau = 0.07, m = 0.999, batchsize = 256$, and $lr = 0.03$.

We can report a top-1 accuracy of 65.1% on DomainNet-Real. It should be noted that a supervised ResNet-50 reaches an accuracy of 63.8%. To test domain generalization, we use the frozen DomainNet-Real trained backbone, and simply feed in a different domain. This set up is possible in DomainNet because all 6 datasets share the same labels. We essentially follow a single source domain generalization protocol Wang et al. (2021). The MoCo-v2 and MoCo-v3 numbers reported in table 2 were evaluated using a leave-one-group-out (LOGO) strategy. The three groups are (clipart, infograph), (painting, quickdraw), (real, sketch) (Yu et al., 2024)

Here, due to differences in evaluation protocols, the comparison is fair only between certain groups. Since we train on Real domain, data points where the Real domain was included in the LOGO training strategy should be considered. Since the LOGO strategy uses two domains to train, we would like to emphasize that our training protocol of Single Source domain generalization is a more difficult task. Hence we can compare the domain generalization of Clipart, Infographic, Painting and QuickDraw domains fairly. In all of these domains, our method outperforms MoCo-v2 and MoCo-v3 by a significant margin. These results demonstrate that symmetric queue-based architectures enable ViTs to learn more transferable representations than asymmetric designs. The consistent improvements across diverse domains indicate that our architectural principles capture fundamental invariances rather than dataset-specific biases.

## 4.3 Stability and Ablation Study

For our stability and ablation study, we chose to focus on ImageNet-100 as the main dataset. While ImageNet-1k serves as a valuable benchmark dataset for SSL, most practical datasets would have 100k to 200k images. Academic researchers, industry practitioners, and domain-specific applications—from medical imaging to satellite imagery—rarely have access to ImageNet-scale data. ImageNet-100 (130K images) better represents this realistic setting while maintaining sufficient complexity for meaningful SSL evaluation. A single run of ImageNet-100 takes around 24-26 hours, which is vital to test reproducibility in a limited compute environment. To provide evidence for stability, we used the same parameter set of $queue = 16384, \tau = 0.07, m = 0.999, batchsize = 256$,

| Queue Size | LP Accuracy | Epochs to convergence | Final contrastive loss |
|---|---|---|---|
| 4096 | 66.79% | 465 | 0.854 |
| 8192 | 67.71% | 464 | 1.08 |
| 16384 | 69.8% | 582 | 1.22 |
| 32768 | 68.96% | 508 | 1.66 |
| 65536 | 68.34 | 512 | 1.83 |

Table 3: Queue sweep and effect on top-1 accuracy, number of epochs needed for convergence and final contrastive loss.

and $lr = 0.03$ for multiple runs and recorded a final downstream accuracy of 69.8%, 69.9%, 69.2% and 69.3%. While it was not possible to have multiple runs over different sets of parameters due to computational and time restrictions, we offer these runs as evidence that our method is stable and reproducible with $mean = 69.742$ and $std = 0.1890$

We systematically evaluated queue sizes from 4,096 to 65,536 samples to determine the optimal negative sampling diversity in Table 3. The results show a peak performance at K=16,384 (69.8% accuracy), with larger queues showing diminishing returns. This suggests an optimal balance between negative diversity and downstream accuracy. Excessively large queues may introduce stale negatives that harm contrastive learning. The consistent convergence across all queue sizes (464-582 epochs) demonstrates the architectural robustness of our symmetric design.

### 4.3.1 PREDICTION HEAD ABLATION

We ran tests over the same model configurations and parameters including and excluding the prediction head on the query encoder. We noticed that the prediction head is incompatible with the architecture, and does not generate a smooth loss curve. Figure 4 plots the graph between the contrastive loss of the setup with and without the predictor head.

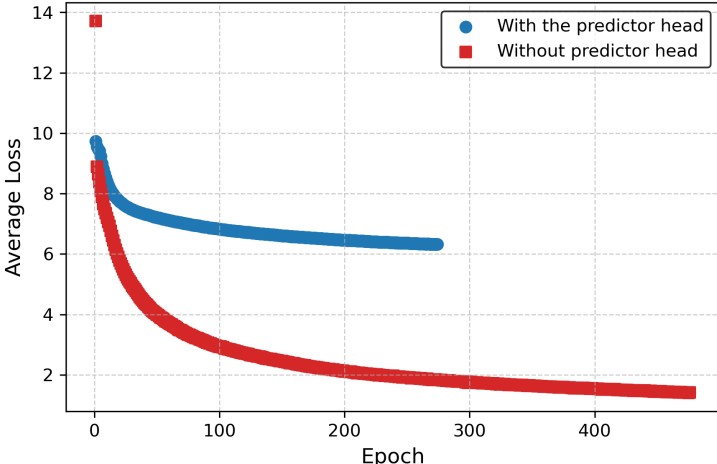

Figure 4: Contrastive loss with and without predictor head. Due to non-decreasing loss of the experiment with the predictor head, the training was stopped early.

We hypothesize that the reason the loss never decreases below a certain threshold is the drift between the query encoder and the key encoder. When the prediction head is applied to in-batch sampling, then no matter the timestep in training, the positive augmentation is always compared against the same negative samples. Hence, even though the query and key encoders essentially form different representation spaces, due to repeated constant consistent comparison, the asymmetry works. However, at different timesteps, a queue is populated with different negative samples from various batches. We hypothesize that since the constant consistent comparison is broken, the contrastive loss does not decrease. This head maps query embeddings into a transformed latent space, while the

key encoder, lacking this head, stores embeddings in a different space. With momentum updates and small batch sizes, this mismatch causes the embeddings in the queue to become stale and incompatible with the current query embeddings, resulting in conflicting gradients and non-convergence. Experimental ablations support this: models with the prediction head fail to converge, while removing it enables stable, low-loss training.

## 5 DISCUSSION

Our evidence suggests that in a ViT setting, queue-based methods and asymmetrical encoder architectures in contrastive learning are fundamentally incompatible. This finding could have potential implications for future SSL designs. The widespread adoption of asymmetrical encoders may have inadvertently discouraged the development of queue-based ViT methods. Our work suggests that queues may have been abandoned because of inherent limits, when an architectural mismatch might have been the true reason.

Current state-of-the-art methods depend heavily on large compute budgets, multi-node clusters and large batch sizes. In contrast, ViT-MoQ achieves competitive performance on a single consumer GPU (RTX 4090, 20 GB VRAM) with a batch size of 512 — a 2–3× reduction in compute requirements compared to prior work. This not only can enable the training of high-quality SSL models in low compute settings and thus expanding the application areas of SSL training, but also allows for a more resource efficient, sustainable training and efficient, smaller architectures.

Our superior domain generalization performance, particularly outperforming MoCo variants by significant margins, suggests that queue-based learning with adapted architecture learns more transferable representations. The diverse negative sampling enabled by queues may force the model to learn more fundamental, domain-invariant features rather than dataset-specific shortcuts. Notably, outperforming supervised ResNet-50 performance (65.1% vs 63.8%) on DomainNet-Real, while dramatically outperforming in cross-domain transfer, indicates our method captures both task-relevant and generalizable features, which is a desirable property for many practical applications.

ViT-MoQ addresses a critical sustainability and accessibility challenge in modern AI research. The 2-3× compute reduction significantly lowers energy consumption and carbon footprint compared to existing transformer SSL methods, contributing to more sustainable AI development. This efficiency enables SSL research in energy-constrained environments, supports researchers at institutions with limited computational resources, and reduces the entry barrier for ViT-SSL.

## 6 CONCLUSION

In this paper, we introduce ViT-MoQ, which establishes the first framework for queue-based resource efficient ViT SSL and demonstrates superior domain generalization under the SSDG protocol. Our key contributions are:

- We show that asymmetrical encoder architectures are incompatible with the momentum queue framework for Vision Transformers. We resolve this incompatibility by adopting symmetrical encoders and show the training to be stable and yielding consistent results.

- We achieve competitive SSL performance (61.3% ImageNet-1K) using 2-3× less compute resources than existing transformer methods. This enables high-quality SSL training on single consumer GPUs with 165 GPU hours total.

- Our approach reduces energy consumption and carbon footprint while democratizing SSL research access. This enables a broader participation from researchers at resource-constrained institutions and supporting more sustainable AI development practices.

- We demonstrate significant improvements in domain generalization under single-source protocols, outperforming MoCo variants by substantial margins (e.g., 44.42% vs 28.4% on painting domain). We also exceed the supervised baseline set on the real domain by ResNet-50 (63.8% vs 65.1%)

Especially the significant increase in domain generalization of ViT-MoQ is strong evidence that the combination of queue-based methods and ViT backbones need to be further explored. With ViT-

MoQ, we demonstrate a first step in this direction and show that this combination is beneficial and warrants further research.

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
