# OpenReview forum: "VIT-MOQ: REVISITING MOMENTUM QUEUES FOR RESOURCE-EFFICIENT VISION TRANSFORMERS AND DOMAIN GENERALIZATION IN SELF- SUPERVISED LEARNING"
_ICLR.cc/2026/Conference — Submitted to ICLR 2026_

### Official Review · Reviewer_oR79 · 2025-10-17

**Soundness:** 2
**Presentation:** 2
**Contribution:** 1
**Rating:** 2
**Confidence:** 4

**Summary:**

ViT-MoQ is a compute-efficient self-supervised learning method for Vision Transformers that reintroduces momentum queues, previously considered obsolete. The key insight is that symmetric encoder architectures are crucial for queue-based learning in ViTs, unlike the asymmetric designs used in recent SSL methods. ViT-MoQ achieves competitive ImageNet-1K performance with only a single GPU and 165 GPU hours, while showing strong domain generalization on DomainNet-Real, outperforming MoCo variants by large margins. This work challenges the belief that momentum queues are outdated in the transformer era and demonstrates that proper architectural design enables their effectiveness. ViT-MoQ makes high-quality SSL accessible on modest hardware and promotes sustainable, green AI research.

**Strengths:**

They provided a clear presentation to explain their method and also gave the difference between their method and previous approaches. And they showed the efficacy and effiency by the relavant experimental results.

**Weaknesses:**

1) From the method level, I think the method's novelty is not enough for ICLR. The topic they studied is only on ViT and partial part of SSL. From my perspective, their framework consists of: a) symmetric encoders and b) a queue. For a), I think it's totally the same as previous work (Grill, Jean-Bastien, et al. "Bootstrap your own latent-a new approach to self-supervised learning." Advances in neural information processing systems 33 (2020): 21271-21284.); for b), I did not see the novelty of just introducing a queue. So they combine a previous work and a normal structure. That's all. They also did not provide enough contents in method part to explain or prove their method is novel.  I admit that the method has some novelty but not enough for ICLR.
2) For the experiments, my concern is that the experiments are not enough. They compared to enough baselines but they just focused on insufficient datasets. Also they used two different ImageNet datasets, which can not support that they performed extensive experiments. From the performance/effiency, for example, in table1, I did not see that how effective their method is. Even if they could decrease the overhead or GPU hours, but they lost the performance. I understand that there is tradeoff but the performance can not convince me that the method is both effective and efficient.
3) Minor point: the presentation is not mature but not my major concern.

**Questions:**

N/A

---

> ### Author Response · Authors · 2025-12-02
>
> Dear reviewer oR79,
> Thank you for your evaluation.
>
> While we agree that simply combining existing components is not novel, we believe that the paper nevertheless fills some gaps in the scientific landscape:
>
> - We demonstrate that using a queue (which many new methods have abandoned) alongside a ViT backbone can significantly enhance the acquisition of global, domain-independent features. To our knowledge, this area has not been adequately explored thus far, and our work highlights an area that warrants further investigation.
>
> - We demonstrate that asymmetric architectures cannot be combined with queues, and that this is due to the projection head. This informs future architectural designs.
>
> We can present new experimental results on smaller ViT variants that more clearly demonstrate which changes are responsible for superior domain generalisation. MoCo-v3 already demonstrates superior domain generalisation to MoCo-v2, suggesting a potential benefit from the ViT backbone alone. However, it appears that the significant enhancement in domain transfer capabilities can only be achieved through the combination of increased negative sample availability in a queue and the advanced representations provided by a ViT backbone over a ResNet50 backbone.
>
> We can also add experiments that provide more evidence that our hypothesis about the problems with asymmetric designs and momentum queues is inherent and stems from a divergence of representations between the encoders in asymmetrical architectures.
>
> We hope that with these additional experiments included and a stronger focus on domain transfer, the novelty is clearer, and this paper would be worth publishing in ICLR.

---

### Official Review · Reviewer_8A9T · 2025-10-27

**Soundness:** 1
**Presentation:** 1
**Contribution:** 1
**Rating:** 0
**Confidence:** 5

**Summary:**

This paper presents a novel self-supervised learning method from images inspired by queue learning methods such as MoCo, with the objective of obtaining strong downstream performance with small training budget and small batch sizes, in particular with single-gpus training. The paper advocates for symmetric encoder architectures as essential to reach this goal. Experimental results are presented on linear probing on imagenet and domain generalization on DomainNet-Real.

**Strengths:**

As shown in Table 2, the presented method offers good performance on domain generalization in comparison to MoCo. There might be something fundamental that makes the method different from MoCo that might be interesting to investigate.

**Weaknesses:**

- The main issue with this paper is that the claims are not supported by the experiments. The main contribution is the new method that is supposed to provide strong performance with low-budget compute. In practice, there is no fair comparison that allows one to conclude that other SSL methods could achieve the same performance as ViT-MoQ with the same amount of compute. Concretely Table 1 shows that ViT-MoQ achieves 61.3% accuracy on ImageNet linear probing with 165 GPU hours. MoCo-v3 with the same ViT-S/16 backbone achieves 72.5% with 614 GPU hours. What about MoCo-v3 with 165 GPU hours ? The same can be said for more recent state-of-the-art methods such as DINO.

- Regarding the argument of ViT-MoQ not requiring large batch sizes, there is no experiment, but the paper mentions that “state-of-the-art methods require substantial computational resources with large batch sizes (4096)”. Large batch sizes are required for accelerating training but it is not a strong limitation and these methods might be trained with lower batch sizes.

- The presentation of the paper is of very low-quality. The paper is very unclear. For example, the authors make the argument that “Symmetric encoder architectures are essential for queue-based learning in ViTs” but they use an asymmetric architecture with a teacher student setup and an exponential moving average weights for the teacher, which is asymmetric. The vocabulary needs to be more precisely defined.

- I don’t understand Figure 3 at all, y-axis is reportedly the accuracy but the values go above 100.

- Figures are blurry, please use .pdf files instead of .png or .jpeg.

- The writing is very poor. A few examples just at the beginning of the introduction: the 1st sentence: “Self-Supervised Learning (SSL) has perceived much attention, especially in the language domain.” is incomplete; many sentences are short, with no connections between them. “often employing clusters of A100 and H100 to achieve these numbers”. No numbers are being referred to, the sentence does not make sense. “Other methods like” is repeated twice in two consecutive sentences. Poor quality writing hinders the understanding of the reader and makes it hard to take the paper seriously.

- Experiments from Table 3 and Figure 4 are not relevant and do not correspond to any claim made in the abstract/intro.

- In this state the paper cannot be accepted and further research is required.  Some advices for the authors:
Try to pick one hypothesis and make convincing experiments that answers this hypothesis. For example if the low-budget compute is the hypothesis, compare several SOTA methods with yours with the same small budget and compare the performance. If the hypothesis is better domain generalization performance, try to focus on this aspect and do ablations to understand which component of your method contributes to how much improvement you observe. Pay close attention to the presentation. In particular the writing and the figures, and how they help support your claims.

**Questions:**

no

---

> ### Author Response · Authors · 2025-12-02
>
> Dear reviewer 8A9T,
> Thank you for your feedback and for identifying the weaknesses. Please find our comments on your points below:
>
> 1. While we cannot train MoCo-v3 from scratch and measure its performance after 165 GPU hours, we can estimate its likely performance after this time based on the paper. After 165 hours, MoCo v3 will have reached approximately 80 epochs. At 80 epochs, MoCo v3 reports a KNN accuracy of 47–48%. While this does not directly equate to linear probe accuracy, it is one of the closest available metrics for comparison. We can include this information in the paper.
>
> 2. We disagree with the assertion that larger batches only accelerate training. Contrastive learning requires a large number of negative examples to sample from. While it is possible to train these methods with low batch sizes, problems with accuracy will arise (as demonstrated by SimCLR through a batch size sweep). To maintain optimal performance in methods such as MoCo v3, DINO or BYOL, large batch sizes are necessary. Earlier methods incorporated queues for this reason, to keep a large number of negative samples, which smaller batches cannot provide.
>
> 3. We do not use a student-teacher approach, which is not mentioned anywhere in the paper when describing our architecture. We use an architecture with two symmetric encoders that are trained in parallel. The encoders are symmetric because they both contain a ViT-S/16 backbone and a projection head, and the key encoder is updated using momentum. This is quite different from a student-teacher approach, in which the student is trained using a fully trained teacher. We will try to clarify this by improving Fig. 1 and its description.
>
> 4. The y-axis shows the accuracy achieved per 1,000 GPU hours, which is a metric that we used to compare the efficiency of multiple models that were trained using the same dataset. As this metric is confusing, Fig. 3 will be removed, as the same information is already contained in Fig. 2.
>
> 5 + 6. We will improve the figures in the final version of the paper and proofread it again in more detail to ensure clarity and improve the writing style.
>
> 7. We apologise if the purpose of these two elements was unclear: Figure 4 is essential to our first claim that reintroducing the momentum queue requires a symmetric architecture because it shows that a predictor head (as in the asymmetric MoCo-v3 architecture) is incompatible with ViT-MoQ training. As reintroducing the queue to a ViT backbone is one of the paper's main contributions, identifying a cause that prevents this integration with the MoCo-v3 architecture is extremely relevant.
> As for the queue size sweep (Table 3), introducing a new parameter, the queue size, is important for the application of the method, as it shows diminishing returns after a certain point for a given dataset, and it also confirms statements from the MoCo-v3 paper. We therefore consider this to be an important contribution, in line with batch size sweeps for the other methods.
> In future, we will include more results from experiments with smaller ViT variants to:
> (a) show a more thorough comparison between different architectures; and
> (b) conduct ablation studies to more clearly identify the cause of the improved domain transfer. This will shift the focus from lower compute requirements to the improved domain transfer capability of our method.
>
> We hope that we have clarified some points that were clearly misunderstood. Please let us know whether the suggested improvements would change your view of our paper.

---

### Official Review · Reviewer_qaEH · 2025-10-28

**Soundness:** 2
**Presentation:** 3
**Contribution:** 2
**Rating:** 2
**Confidence:** 4

**Summary:**

The paper introduces a self-supervised contrastive learning method applied to a ViT-S/16 architecture. Its main innovation lies in reintroducing the momentum queue mechanism from MoCo-v2---originally developed for ResNet-50 architectures---while removing the prediction head commonly used in recent self-supervised learning (SSL) approaches. The authors argue that this modification is crucial to successfully adapting momentum queues to transformer-based architectures.  The study focuses on the low-compute regime, where the batch size is below 256, enabling training on a single GPU. Despite this resource constraint, the proposed approach achieves 61.3% top-1 accuracy on ImageNet-1K using 165 GPU hours, compared to 72.5% obtained by MoCo-v3 after 614 GPU hours. Moreover, in domain generalization tasks, the method demonstrates substantial improvements over MoCo-v3, suggesting that the proposed design leads to more transferable and robust representations.

**Strengths:**

- the paper is relatively clear.
  - It addresses self-supervised learning from a low-compute perspective, which
    is a timely topic given the increasing resource demands of large-scale
    foundation model development.
  - the results on domain generalization are interesting.

**Weaknesses:**

- The main contribution of the paper is the empirical assertion that a
    symmetric architecture is necessary for learning with momentum queues---i.e.,
    that prediction heads on the query encoder should be removed in such
    setups. However, the empirical support for this claim is limited: it is
    evaluated only on a single architecture (ViT-S/16), and alternative
    explanations, such as hyperparameter choices, could account for the results
    shown in Figure 4. The experiment would be more convincing if supported by
    a clear theoretical reason, but the explanations provided are currently
    rather vague.
- The reported performance is not particularly compelling and lacks
    comparisons to existing SSL baselines in the low-compute regime. Examples
    of relevant baselines include:
       (i) MoCo-v3 (or other SSL methods) with early stopping: Instead of training the baseline model for the full epoch budget, it would be informative to see how performance evolves under reduced training budgets, and how many epochs are required to match the proposed approach on ImageNet.
       (ii) Smaller architectures trained with MoCo-v3: For instance, in Wang et al., A Closer Look at Self-Supervised Lightweight Vision Transformers (ICML 2023), ViT-Tiny models with only 5.7M parameters achieve strong performance. This suggests that small architectures with standard SSL methods can be competitive (see also references in their paper).
Including these comparisons would provide a more rigorous evaluation of the proposed approach.

- The code is not available at the time of submission, although the authors indicate it will be released later.

**Questions:**

Suggestion 1: there are several claims about Green AI in the paper that seem
  quite exagerated.  The footprint of a model is not only about computational
  time for training, it is also about the cost at inference time (which is far
  from negligible here), its lifespan (for how long the model will be used
  before being replaced), how much it will be used and for what. By taking all
  of these aspects into account, it is far from clear that the development of a
  new VIT-S/16 (which is probably worse than Moco-V3, DINO-v2/v3 on many
  downstream tasks) will help achieving greener AI practices.

  Suggestion 2: if light-weight self supervised learning is the main objective,
  lighter architectures than Vit-S/16 would help making the case more convincing
  (ViT-tiny for instance).

---

> ### Author Response · Authors · 2025-12-02
>
> Dear reviewer qaEH,
>
> Thank you for your valuable feedback and questions. We would like to address your queries and identified weaknesses:
>
> Suggestion 1: The inference costs for models that use the same backbone are actually comparable, since the inference costs for a ViT backbone are mostly determined by the backbone architecture. Therefore, you are correct in saying that we only reduce the training costs. We will place lower importance on the GreenAI aspect and focus more on the improved domain generalization capabilities.
>
> Suggestion 2: We are currently conducting experiments with small ViT variants to determine which component of ViT-MoQ is responsible for the enhanced domain transfer capability. Since this will also include training time measurements for different combinations (queue versus different batch sizes), we hope that this will satisfy your request.
>
> Weakness 1: You are right; the current evidence is based on comments in the original MoCo-v3 paper about why the queue was abandoned, as well as our own difficulties in training ViT-MoQ in an asynchronous setup (see Fig. 4). We will add different ViT architectures to rigorously test our hypothesis further and demonstrate that the problem occurs in any setup (independent of hyperparameter settings). We are currently experimenting to provide additional evidence for the hypothesis that, in an asynchronous setup, diverging encoder representations lead to instability.
>
> Weakness 2: The goal of this method is not to outperform state-of-the-art (SOTA) methods, but rather to train supervised self-supervised learning (SSL) more efficiently within a given compute budget. However, most established models do not report compute budgets using a standard metric, which makes it difficult to compare them without training all the models ourselves. This prevents us from making a fair comparison with other methods such as VICReg, TriBYOL or FastSiam due to resource limitations. However, we can include comparisons with smaller models, as suggested (see Suggestion 2), where we fix the compute budget (number of epochs or GPU hours) and also measure compute budget to achieve the same performance. We can also include estimates for MoCo-v3 with the same compute budget based on the K-nearest neighbour performance over epochs reported in the paper.
>
> We will focus more on analysing domain generalisation as a contribution of the paper. We hope that this different focus, along with additional experiments on smaller ViT variants (if the evidence supports our initial claims), would be enough to change your evaluation of our paper.

---

### Official Review · Reviewer_dbwY · 2025-11-03

**Soundness:** 2
**Presentation:** 1
**Contribution:** 2
**Rating:** 4
**Confidence:** 3

**Summary:**

The paper proposes ViT-MoQ, a compute-efficient contrastive SSL method. Its core mechanism is the reintroduction of the momentum queue, a concept popularized by MoCo but later phased out in favor of purely large-batch or asymmetric-encoder methods.

One of the paper central claim is that symmetric encoder architectures are crucial for the queue-based mechanism to function effectively with ViTs. By employing the momentum queue, ViT-MoQ achieves comparable performance while reducing the required batch size and computational footprint. Furthermore, the method is claimed to be particularly effective on the downstream task of Domain Generalization (DG), where models must perform well on unseen domains.

**Strengths:**

The primary strength of ViT-MoQ is its commitment to resource efficiency. The paper offers a vital pathway for researchers with limited computational resources to engage with a ViT-based SSL, thus promoting democratisation of research. The paper promises to share the code.

The idea of the momentum queue itself is not new. The paper's novelty rests entirely on its efficient integration with ViT under the constraint of symmetry. Note, I feel like that claims are a bit overstated, like ``asymmetrical encoder architectures are incompatible with the momentum queue framework for Vision Transformers''. This claim is based on experiments that just show this in a specific case.

**Weaknesses:**

I am not convinced by the interest of being able to training on a commodity machine. Since we can obtain much stronger ViT-S model for a regular cost, I do not see a scenario when one would like to train from scratch a model, while it is possible to simply use a foundational model of limited size yet trained with more compute.

- domain generalization is presented on small benchmarks, and somehow this kind of generalization seems almost anachronical considering the open-vocabulary setting that has emerged in recent years, notably those resulting from vision-language models.
- presentation is poor, with figures that needs to be re-generate: I cannot  barely read the text in Figure 2 and 3.
- In Figure there is only one label x-axis point, so we don't even know if it is regular or log-scale graduation.

**Questions:**

Could you please elaborate a bit more on the experimental conditions for the baseline techniques compared to "ours" (Moco-v2 -v3), and possibly comment on how they differ. It is not clear to me whether there could be some cofounding factor at play here.

---

> ### Author Response · Authors · 2025-12-02
>
> Dear reviewer dbwY,
>
> Thank you for your reply! We will, of course, improve the presentation and redraw the graphs for better readability.
>
> With regard to the first weakness you mention, open-vocabulary methods approach domain generalisation from a different direction (by training on as many domains as possible to solve the problem by increasing the amount of training data). However, for us, domain generalization is a benchmark for analysing the generalizability of the features extracted by the model. If a model can extract more general features, it can adapt faster to previously unknown domains. In our view, a method that is capable of automatically extracting domain-independent features (without the need to use all the necessary domains in training) would always be preferable and would even allow for smaller, more efficient open-vocabulary models.
>
> We are already conducting experiments to show which feature of the model (queue, ViT backbone) is responsible for the increased domain transfer abilities. We hope this wiill be enough to change your evaluation

---

### Meta-Review · Area_Chair_sifo · 2026-01-06

**Summary:**

There are three consistent issues raised by multiple reviewers
1. Novelty and contribution by all reviewers
2. Experimental Rigor (Reviewers qaEH, 8A9T, oR79)
3. Presentation Quality (Reviewers dbwY, 8A9T)

These factors determined the decision for the paper

**Reviewer Concerns:**

The rebuttal really doesn't not provide any new experimental results and the authors acknowledge the issues with presentation and figures. Hence, I feel that none of the issues were clearly addressed

**Reviewer Scores:**

Since the issues were not addressed, the scores likely would remain the same.

---

### Decision · Program_Chairs · 2026-01-26

Reject